# The Role of Aquaporins in Spinal Cord Injury

**DOI:** 10.3390/cells12131701

**Published:** 2023-06-23

**Authors:** Terese A. Garcia, Carrie R. Jonak, Devin K. Binder

**Affiliations:** 1Division of Biomedical Sciences, School of Medicine, University of California, Riverside, CA 92521, USA; 2Center for Glial-Neuronal Interactions, University of California, Riverside, CA 92521, USA; 3Neuroscience Graduate Program, University of California, Riverside, CA 92521, USA

**Keywords:** astrocytes, aquaporin, aquaporin-4, spinal cord injury, edema

## Abstract

Edema formation following traumatic spinal cord injury (SCI) exacerbates secondary injury, and the severity of edema correlates with worse neurological outcome in human patients. To date, there are no effective treatments to directly resolve edema within the spinal cord. The aquaporin-4 (AQP4) water channel is found on plasma membranes of astrocytic endfeet in direct contact with blood vessels, the glia limitans in contact with the cerebrospinal fluid, and ependyma around the central canal. Local expression at these tissue–fluid interfaces allows AQP4 channels to play an important role in the bidirectional regulation of water homeostasis under normal conditions and following trauma. In this review, we consider the available evidence regarding the potential role of AQP4 in edema after SCI. Although more work remains to be carried out, the overall evidence indicates a critical role for AQP4 channels in edema formation and resolution following SCI and the therapeutic potential of AQP4 modulation in edema resolution and functional recovery. Further work to elucidate the expression and subcellular localization of AQP4 during specific phases after SCI will inform the therapeutic modulation of AQP4 for the optimization of histological and neurological outcomes.

## 1. Introduction

Traumatic spinal cord injury (SCI) is a debilitating and life-altering condition [1,2]. SCI can lead to loss of motor, sensory, and autonomic function [3,4,5]. These impairments can be additionally linked to comorbidities that introduce additional obstacles for SCI patients in their daily lives [6,7]. For example, psychological comorbidities of SCI that have recently received a great deal of attention include depression, anxiety, and post-traumatic stress disorder [8]. The initial mechanical insult to the spinal cord (termed primary injury) is followed by a cascade of events that may exacerbate the injury (termed secondary injury). Histopathological processes occurring in the injured spinal cord include breakdown of the blood–spinal cord barrier (BSCB), edema (water accumulation), hemorrhage, inflammation, excitotoxicity, cavitation, and glial scar formation. One major goal of SCI research is to identify potentially reversible causes of secondary injury.

Animal models of SCI include contusion, compression, and transection [9]. Contusion injuries involve impact to the exposed spinal cord with an impactor or a weight drop system [10,11,12]. The most common model involves creating an injury with a computer-controlled impactor (Infinite Horizons) which creates user-selectable force levels (30–300 kilodynes, i.e., 0.3–3 N) and allows the impact tip dwell time to be modulated. Real-time probe force and displacement graphs provide the operator with a precise detailed record of the kinematics of each impact. These parameters can be modified to generate mild, moderate, or severe SCI. Compression injuries are created by constricting the exposed cord with clips, forceps, or weight application [13,14,15]. Transection models involve durotomy (opening of the dura) and myelotomy (incision into the spinal cord parenchyma, such as complete transection or hemisection of the cord). In general, contusion models more accurately model typical human SCI [16,17,18].

## 2. Spinal Cord Edema

After acute SCI, edema develops at the injury site and spreads rostrally and caudally, leading to increased cord water content and cord tissue volume [19]. Edema is generally divided into *vasogenic* and *cytotoxic*, but there may be a combination of the two. *Vasogenic* edema results from breakdown of the BSCB, resulting in an accumulation of fluid in the interstitial (extracellular) space of the cord. *Cytotoxic* edema results from fluid flow into astrocytes resulting in cellular (hence “cytotoxic”) swelling. Contusion SCI results in traumatic BSCB disruption, and hence is thought to be largely a model of vasogenic edema; in contrast, compression SCI causes ischemia to cord vessels and swelling of astrocytes, and hence is thought to be largely a model of cytotoxic edema. It is clear that injury severity correlates with the development of edema at the injury site [20], and the extent of edema impacts secondary damage and neurological outcome following injury [19,21,22,23].

What is the mechanism by which edema contributes to secondary injury? Studies in both rodent SCI models and in humans have now clearly indicated that spinal cord edema at the injury site leads to the swelling of the cord, compression of the cord against the surrounding dura within the spinal canal, an increase in intraparenchymal cord pressure, ischemia, and eventual infarction [24,25,26]. Thus, edema can essentially lead to or exacerbate a spinal cord “stroke”. Vascular disruption and BSCB breakdown contribute to fluid, protein, and blood accumulation within the parenchyma of the cord, further exacerbating edema [27,28]. In a rat model of thoracic contusion SCI, spinal cord edema (as assessed by measurement of water content at the lesion site at multiple time points) was found to occur by 6 h and to peak at 72 h [29]. Interestingly, the BSCB is thought to be repaired by 2–3 weeks after trauma [27], suggesting that the acute period (at least through 72 h post-injury) may represent a “therapeutic window” for intervention to alleviate edema and mitigate secondary injury.

Recent human studies of SCI have extensively demonstrated the therapeutic importance of edema reduction in functional outcome. What is apparent from a review of the neurosurgical literature on SCI is that decompressive laminectomy, duraplasty, and even myelotomy do not fully relieve edema or local pressure in the spinal cord. The key studies in this regard have been conducted as part of the ISCoPE (Injured Spinal Cord Pressure Evaluation) trials. These are the first studies in humans to evaluate intraparenchymal spinal cord pressure. These studies find massive increases in intraspinal pressure at the injury site that falls off farther away, indicating that peak edema at the injury site is likely responsible for secondary injury [30,31,32,33,34]; indeed, intraspinal pressure and spinal cord perfusion pressure predict neurological outcome [33].

The above studies establish the importance of edema within the cord contributing to secondary injury. Ample clinical data over decades with the use of methylprednisolone [35] have indicated minimal salutary effects of intravenous methylprednisolone on reducing cord edema or improving outcome [36]. As more is being uncovered about the mechanisms underlying edema formation following injury to the central nervous system (CNS), interventions aimed at this issue are becoming more clinically important. To date, there is no effective treatment for edema following SCI. Thus, the discovery and characterization of molecular water channels (the aquaporins) and their examination in the context of SCI are of critical scientific and translational importance.

## 3. Aquaporins

Aquaporins (AQPs) are a family of membrane proteins that function as water channels in many cell types and tissues in which fluid transport is crucial [37,38,39,40]. AQPs are small hydrophobic integral membrane proteins (~30 kDa monomer) that facilitate bidirectional water transport in response to osmotic gradients [41]. Multiple mammalian aquaporins have been identified, including AQP0, AQP1, AQP2, AQP4, and AQP5, which transport water only (aquaporins), and AQP3, AQP7, and AQP9, which also transport glycerol (aquaglyceroporins) [39,41].

Aquaporin-4 (AQP4) [42,43,44,45] (Figure 1) is of particular interest in neuroscience as it is expressed in the brain and spinal cord by astrocytes and ependymal cells, especially at specialized membrane domains including astrocytic endfeet in contact with blood vessels and astrocyte membranes that ensheathe glutamatergic synapses [40,46,47,48,49]. AQP9 is also expressed in astrocytes at low levels, and AQP1 is expressed in choroid plexus epithelial cells (Figure 1C, right inset). AQP4 has two major isoforms (AQP4-M1 and AQP4-M23). AQP4-M23, but not AQP1-M1, forms square arrays in the astrocyte plasma membrane, known as orthogonal arrays of particles (OAPs) [50]. OAPs can be seen in freeze-fracture electron micrographs [51,52,53,54,55]. It has been demonstrated that activity-induced water fluxes in the neocortex may represent water movement via aquaporin channels in response to physiological activity [56,57].

In the spinal cord, AQP4 expression is found ubiquitously in both gray and white matter and is prominent not only in astrocytic endfeet but also at the glia limitans, where the spinal cord directly contacts cerebrospinal fluid [49] (Figure 2). This pattern of expression suggests that AQP4 may play a crucial role in spinal cord water homeostasis.

## 4. Aquaporin 4 Expression after SCI

The first and still most comprehensive study of AQP4 changes in rodent SCI was published in 2006 [58]. The authors demonstrated the robust baseline expression of AQP4 in gray and white matter astrocytes, specifically in astrocytic processes surrounding neurons and blood vessels. This immunoreactivity was particularly strong in the glia limitans externa and interna. These findings in the baseline rat spinal cord were very similar to those published in 2004 in the mouse spinal cord [49]. Indeed, the authors commented that the spatial distribution of AQP4 suggests the critical role that astrocytes expressing AQP4 play in the transport of water from blood/cerebrospinal fluid to spinal cord parenchyma and vice versa. Following thoracic contusion SCI, biphasic changes in astrocytic AQP4 levels were observed, including early downregulation and subsequent persistent upregulation. This early downregulation occurred in the acute and subacute periods following injury (up to 14 days) and subsequently there was increased AQP4 expression in both gray and white matter of the spinal cord (day 21 to 11 months post-injury) [58].

Interestingly, this group also studied neurological outcome based on injury severity (mild, moderate, and severe) and correlation to AQP4 expression. At 3 weeks post-injury, animals showing more impairment on the Basso, Beattie, and Bresnahan locomotor scale [59] had lower levels of AQP4 protein. The authors suggest that changes in expression and/or localization of AQP4 (and its anchoring proteins dystrophin and α-syntrophin) may lead to impaired water clearance in the chronically injured cord [58,60].

In a human study, several spinal cord tissue samples were analyzed for the expression of AQP4. Tissue was obtained at autopsy from individuals that had been injured for a duration of 1–2 years (so of course no data were available from human tissue in the acute/subacute period). The authors found that human SCI tissue samples contained GFAP-positive astrocytes both with and without AQP4 labeling at the lesion epicenter [60]. In contrast to the relative lack of expression of AQP4 in astrocytes at the lesion epicenter, AQP4 labeling was increased in spared white matter compared to uninjured cord tissue [60].

Subsequently, other studies using different SCI models and species have also indicated chronic increases in AQP4 expression after SCI [61,62,63] or AQP4 modulation by pharmacological agents after SCI [25,64,65,66,67,68,69,70,71,72,73,74]. These studies have previously been reviewed [75,76]. Overall, the studies suggest that there is a long-term upregulation of AQP4 in the chronically-injured spinal cord, but whether this is compensatory or a result of reactive astrocytosis is unclear.

## 5. SCI Phenotype in AQP4^−/−^ Mice

The above studies of AQP4 regulation in rodent and human tissue strongly suggest that there is a functional role for AQP4 in spinal cord water homeostasis, and hence, the extent of edema following SCI. However, gene expression data and pharmacological modulation data do not unambiguously determine the functional role of AQP4 in SCI. Therefore, several studies have been conducted to test the role of AQP4 in SCI by using AQP4^−/−^ mice.

AQP4^−/−^ mice were originally generated by targeted gene disruption in 1997 [77], and recently an astroglial conditional deletion of AQP4 has been generated [78]. AQP4^−/−^ mice and astroglial conditional AQP4 knockouts are grossly normal phenotypically, do not manifest overt neurological abnormalities, altered blood–brain barrier properties, abnormal baseline intracranial pressure, impaired osmoregulation, or obvious brain dysmorphology [79,80,81].

In vivo studies of these mice demonstrated a functional role for AQP4 in brain water transport. AQP4^−/−^ mice have markedly decreased accumulation of brain water (cerebral edema) following water intoxication or focal cerebral ischemia [82] and impaired clearance of brain water in models of vasogenic edema [81]. Clearance of seizure-induced edema may also be AQP4-dependent [83]. Impaired water flux into (in the case of cytotoxic edema) and out of (in the case of vasogenic edema) the brain makes sense based on the bidirectional nature of water flux across the AQP4 plasma membrane channel at the blood–brain barrier. The recently-generated astroglial-conditional AQP4 knockout mouse line demonstrates a 31% reduction in brain water uptake after systemic hypoosmotic stress [78]. Similarly, mice deficient in dystrophin or α-syntrophin, in which there is mislocalization of AQP4 protein [84,85,86], show attenuated cerebral edema in response to hypoosmotic stress [85,87].

Based on the brain studies above, one would expect similar results in the spinal cord of AQP4^−/−^ mice; in other words, amelioration with AQP4 deletion in models of cytotoxic edema and exacerbation in models of vasogenic edema. That is exactly what has been observed.

In 2008, Saadoun et al. reported the phenotype of AQP4^−/−^ mice subjected to thoracic compression injury [88]. This is thought to be an ischemic injury leading to cytotoxic edema. The authors found that 2 days after thoracic compression injury, AQP4^−/−^ mice had improved neuronal survival and myelin sparing along with reduced tissue water content and intraparenchymal cord pressure compared to wild-type mice. In addition, AQP4^−/−^ mice showed improved functional outcome in the Basso mouse locomotor scale (BMS) [89], inclined plane test, and footprint analysis. Lastly, AQP4^−/−^ mice showed improved sensory outcome as measured by spinal-somatosensory-evoked responses. Thus, AQP4^−/−^ mice have improved outcome in this model of cytotoxic spinal cord edema.

In 2010, Kimura et al. reported the phenotype of AQP4^−/−^ mice subjected to thoracic contusion injury [90]. Contusion is thought to be an injury leading to the breakdown of the BSCB and vasogenic edema. The authors found that AQP4^−/−^ mice showed greater levels of demyelination, cyst formation, and greater neuronal loss 42 days following thoracic contusion injury (Figure 3). In human and rat models of SCI, removal of necrotic tissue by phagocytic cells at the lesion epicenter typically leads to the formation of fluid-filled cavities/cysts in their place, regardless of the size of the species [91]. These results are interesting, as mouse models of SCI do not typically display cyst formation. In addition, Kimura et al. [89] showed that injured tissue from AQP4^−/−^ mice contained increased water content compared to controls at 2 and 4 weeks following injury. Lastly, AQP4^−/−^ mice showed reduced functional recovery in both the BMS and footprint analysis tasks at 42 days post-injury (Figure 4). In contrast to cytotoxic edema produced by compression injury, contusion injury results in vasogenic edema and disruption of the blood spinal cord barrier, not cytotoxic cell swelling. Thus, AQP4^−/−^ mice have worsened outcome in this model of vasogenic spinal cord edema. From this, it appears that AQP4 expression is required for clearance of vasogenic edema and plays a protective role following contusion SCI.

In 2014, Wu et al. reported the phenotype of AQP4^−/−^ mice after rubrospinal tract hemisection [92]. Transection is thought to be an injury leading to the breakdown of the BSCB and vasogenic edema. AQP4^−/−^ mice showed increased spinal cord edema at 72 h after hemisection compared to wild-type littermates. In addition, there was reduced migration of astrocytes to the lesion (at week 1); greater lesion volume, glial scar formation, and cyst volume (at week 6); and increased retrograde axonal degeneration. This study is in agreement with the protective role of AQP4 following vasogenic edema [92].

## 6. AQP4 Modulation in SCI

The studies in AQP4^−/−^ mice together with the clinical data (summarized above) indicating the critical importance of edema in secondary injury after SCI strongly support the concept of modulation of AQP4 as a therapeutic strategy early after SCI. Studies of spinal cord water content changes in the rat model of thoracic SCI indicate that approximately between 6 h and 72 h (3 days) after acute injury there may be a “therapeutic window” to ameliorate edema and improve outcome. Unfortunately, despite intense effort, selective inhibitors or activators of AQP4 have been difficult to develop [93,94].

Nevertheless, several studies have attempted to target AQP4 and related molecules for therapeutic effect. In 2016, Zhang et al. found that, in mice, the antioxidant astaxanthin alleviated cerebral edema after controlled cortical impact TBI by reducing NKCC1, AQP4 mRNA, and protein levels [95]. In 2018, in a thoracic contusion model, Yan et al. found that pretreatment of rats with inhibitors of AQP4 or NKCC1 attenuated edema and tissue damage after SCI [96]. These studies, while interesting, did not selectively target AQP4 nor assess subcellular distribution of AQP4 or channel activities.

In 2020, a landmark paper was published in *Cell* targeting AQP4 subcellular localization to treat edema after SCI [97]. The authors demonstrated that (1) AQP4 cell-surface abundance increases in response to hypoxia-induced cell swelling in a calmodulin-dependent manner; (2) calmodulin directly binds the AQP4 carboxyl terminus, causing a conformational change and driving cell surface localization; and (3) inhibition of calmodulin in a rat SCI model with trifluoperazine inhibited AQP4 localization to the BSCB, abrogated CNS edema, and accelerated functional recovery compared with untreated animals. The SCI model used in this study was a dorsal column crush injury in 6–8-week-old rats. This would be expected to lead to cytotoxic edema (as a compression model). Hence, inhibition of AQP4 surface localization by the calmodulin inhibitor trifluoperazine was efficacious at limiting post-injury edema and improving neurological outcome.

These findings lead to the novel therapeutic strategy of modulation of AQP4 subcellular localization to treat CNS edema [42]. Given the difficulties in developing pore-blocking AQP4 inhibitors, targeting AQP4 subcellular localization opens up new treatment avenues for CNS edema. In particular, the inhibition of AQP4 membrane localization (as with trifluoperazine) is a novel treatment option for cytotoxic edema when it is desirable to limit AQP4 activity. This may have implications for the treatment of conditions other than CNS edema as well. For example, there is prominent AQP4 subcellular mislocalization in epilepsy [98], for which renormalization of its subcellular localization has also been suggested as a therapeutic strategy [99].

## 7. AQP4, Glial Scar Formation, and Neuroinflammation

Glial scar formation and neuroinflammation are other key processes after CNS injury. In the context of SCI, glial scar formation around the injury involves migration of reactive astrocytes to the site of injury. What role might AQP4 play in SCI-associated glial scar formation and associated neuroinflammatory processes?

Several studies have indicated the involvement of AQP4 in glial scar formation. Interestingly, in AQP4^−/−^ mice, astrocytes demonstrate impaired migration and wound healing rate compared with wild-type mice [100]. AQP4 is polarized to the leading edge of the plasma membrane in migrating wild-type astroglia, suggesting that AQP4 facilitates water influx across the leading edge of a migrating cell [100]. Glial scar formation was impaired in AQP4^−/−^ mice after cortical stab injury [100]. Impaired astrocyte migration toward a cervical spinal cord transection injury was also observed in AQP4^−/−^ mice [92]. This was associated with more severe atrophy and loss of axotomized rubrospinal neurons [92]. These authors concluded that AQP4 not only promotes edema clearance (as shown by Kimura et al. [90]) but also glial scar formation after SCI [92]. Glial scar formation was also reduced after TBI in mice with AQP4 gene silencing by lentiviral vector-delivered small hairpin RNA [101] and in AQP4^−/−^ mice [102].

In addition, there is some evidence that AQP4 deletion affects neuroinflammation [103,104]. For example, neuroinflammation was greatly reduced in AQP4^−/−^ mice in various models (active immunization experimental allergic encephalomyelitis (EAE), adoptive-transfer EAE, and intracerebral lipopolysaccharide injection) [105]. Pharmacologic inhibition of AQP4 with TGN-020 has also modulated inflammation in various models [106,107]. However, the precise pathways by which AQP4 modulates neuroinflammatory cascades remain obscure.

## 8. Summary and Future Directions

The role and importance of AQP4 following SCI is still not completely understood or fully explored. In addition, there remains a lack of mechanistic explanation of AQP4 regulation following injury (mRNA expression, protein expression, and subcellular targeting/distribution). As discussed above, the lack of availability of selective aquaporin inhibitors and activators has also significantly limited research approaches. Overall, however, the evidence indicates that AQP4 water channels play critical roles in the formation and resolution of distinct forms of edema following SCI. In particular, AQP4 water channels, as in the brain, play a protective role in contusion or transection SCI (vasogenic edema), but a deleterious role in compression SCI (cytotoxic edema). These and other considerations lead to a variety of key future directions:▪ Further study should be pursued to more fully characterize the extent of vasogenic and/or cytotoxic edema in different animal models. These results would help to elucidate conflicting findings between studies, both within and across injury models. In the end, understanding the precise role of AQP4 following a wide range of injuries will help to better inform the timing of treatments directed at AQP4 modulation for optimal therapeutic benefit and enhanced neurological outcome.▪ In the same vein, a greater understanding of the mechanisms and timing of AQP4 regulation after SCI is needed. This would include regulation both at the mRNA level as well as the subcellular targeting of the AQP4 protein.▪ Are there distinct roles of AQP4 isoforms M1-AQP4 and M23-AQP4 after injury? This is a largely unexplored question. It is known that the M23-AQP4 isoform is critical to forming membrane OAPs (orthogonal arrays of particles), and an M23-AQP4-null mouse has been generated demonstrating massive depletion of brain AQP4 [108,109]. However, the relative roles of M1-AQP4 and M23-AQP4 after SCI remain to be explored.▪ Test AQP4 modulation approaches in more detail. A *Cell* paper made it clear that, rather than pharmacologically targeting the AQP4 membrane pore itself, therapeutic manipulation could be accomplished by modulating the subcellular targeting of AQP4 [97]. Thus, further research into understanding the subcellular targeting and regulation [110,111] of AQP4 will be helpful both to develop new inhibition as well as activation strategies depending on the edema and injury context. And of course, inhibition of AQP4 membrane localization may help to limit cytotoxic edema, but what would be the analogous approach to *improve* AQP4 expression and membrane targeting to limit vasogenic edema (by increasing AQP4-mediated vasogenic edema clearance)?▪ Elucidate AQP4 molecular partners. AQP4 has been shown to be associated with other membrane channels, such as the inwardly-rectifying potassium channel Kir4.1 [47,112], the mechanosensitive cation channel TRPV4 [113], and the ABC protein/TRP channel complex SUR1-TRPM4 [114]. Further understanding of these interactions may also lead to novel mechanistic interventions. For example, TRPM4 knockout blocked astrocyte swelling in a mouse cerebellar cold injury model [114], and the SUR1-TRPM4 inhibitor glyburide inhibits cerebral edema [115]. Such interactions and mechanisms should be targeted specifically for development of novel SCI therapies.▪ Even if the optimal AQP4 modulatory drug(s) are identified, drug delivery considerations should be considered. How will a given drug access and penetrate spinal cord tissue when it is injured, edematous, and ischemic? It is notable in this regard that in the *Cell* paper [97], the drug (trifluoperazine) was injected directly into the lesion site. Efficacy in preclinical trials must be balanced with assessment of drug concentration at the target tissue.▪ Understand how therapeutic intervention to relieve edema impacts intraspinal pressure, spinal cord blood flow, and neurological outcome. The iSCoPE trials clearly indicated the key importance of ISP in neurological outcome [30,31,32,33,34]. This has now been shown in animal models as well and serves as a key target for therapeutic intervention [116,117]. But there has yet to be a study targeting edema to clearly demonstrate all of the links between relieving edema, improving intraspinal pressure, improving spinal cord perfusion, and improving neurological outcome.▪ Osmotic removal of edema fluid. It has been demonstrated that, through establishing an external osmotic gradient, water can be removed from the brain in a controlled manner under normal and pathological brain swelling conditions. Such an “osmotic treatment device” (OTD) was able to reduce brain tissue water content and improve neurological outcome in mouse models of cytotoxic edema and traumatic brain injury without causing histological damage [118,119,120]. These results established proof-of-principle for the concept of direct osmotherapy for the treatment of CNS edema. It has been hypothesized that a similar OTD placed on the dura mater of the spinal cord at the site of injury can withdraw excess water from the cord parenchyma and thus ameliorate SCI edema and improve vascular perfusion and neurological outcome [29]. Proof-of-principle for this concept could potentially be combined with local drug delivery (for example, AQP4 activators or inhibitors depending on the type of edema). 

Ideally, an integrated strategy for the treatment of SCI-related edema should involve (1) microstructural considerations of tuning AQP4 expression and distribution for the restoration of local water transport/homeostasis at the lesion site; and (2) macrostructural considerations of macroscopic tissue edema and pressure. For example, one can envision a combination of AQP4 modulatory approaches together with macroscopic edema removal via OTD to optimally treat post-traumatic edema. Molecular “optimization” of AQP4 would allow for rapid osmotic equilibration within injured tissue, then the external OTD could gently remove excess edema fluid at the surface, thus decreasing intraspinal pressure and restoring blood flow and spinal cord homeostasis. Such approaches could marry creative control of astrocyte water channel physiology to bioengineering approaches. These considerations lead to entirely novel concepts for limiting edema and secondary damage following traumatic SCI.

## Figures and Tables

**Figure 1 cells-12-01701-f001:**
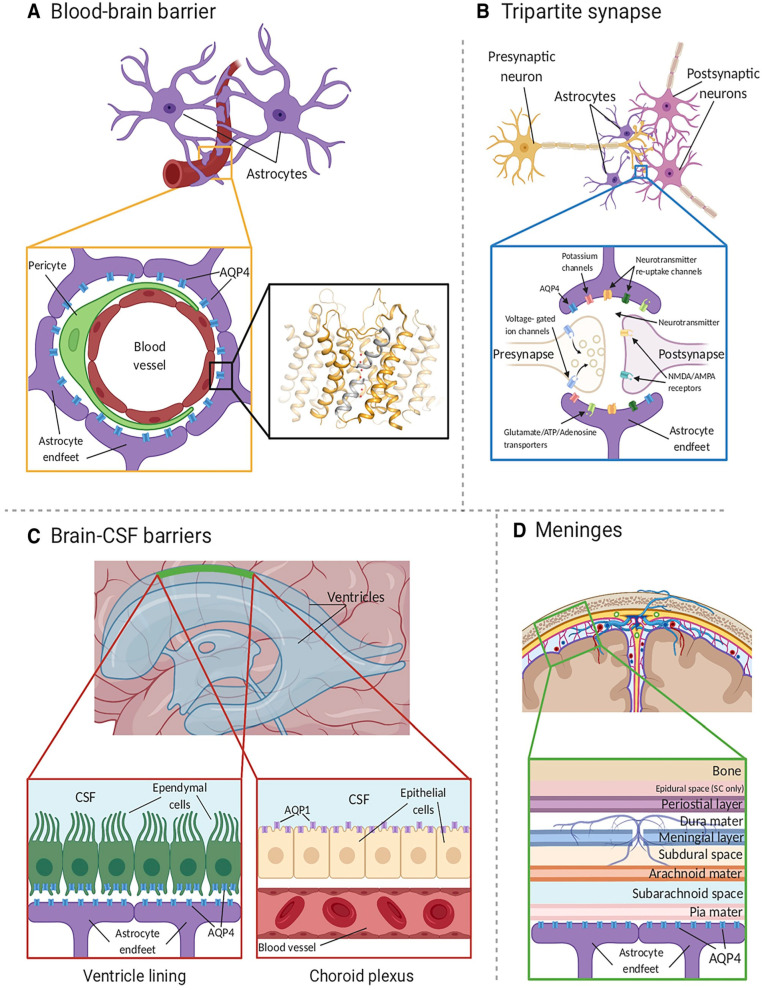
AQP4 localization in the CNS. (**A**) AQP4 (blue) is located within astrocyte endfeet processes surrounding blood vessels in both brain tissue and the BBB. The inset shows the crystal structure of human AQP4 (PDB code 3GD8). AQP4 assembles as a tetramer with each monomer comprising six transmembrane helices and two half-helices (grey). The two half-helices harbor the aromatic-arginine (ar/R) motif that functions as a selectivity filter. Within the pore, water molecules (red spheres) align in a single file. (**B**) AQP4 is also localized at the astrocyte component of the tripartite synapse. During neurotransmission, neurons release mediators and neurotransmitters from synaptic nerve terminals (affecter cells) into the synaptic cleft to communicate with other neurons (effector cells). This synaptic activity induces an increase in intracellular Ca^2+^ concentration, which is accompanied by changed water and solute concentrations in astrocytes, leading to the release of glutamate and other gliotransmitters. This gliotransmission results in feedback to the presynaptic neurons to modulate neuro-transmission. AQP4 plays an essential role in maintaining water homeostasis during this process. (**C**) In ventricles, AQPs are present within ependymal cells lining the brain–CSF interfaces (left inset). AQP4 is localized to the basolateral membrane of ependymal cells and the endfeet of contacting astrocytes. AQP1 (purple) is localized to the apical membrane of the choroid plexus epithelium (right inset). (**D**) CSF within the subarachnoid and cisternal spaces flows into the brain specifically via periarterial spaces and then exchanges with brain interstitial fluid facilitated by AQP4 water channels that are positioned within perivascular astrocyte endfoot processes. Figure and figure legend adopted from [42] with permission.

**Figure 2 cells-12-01701-f002:**
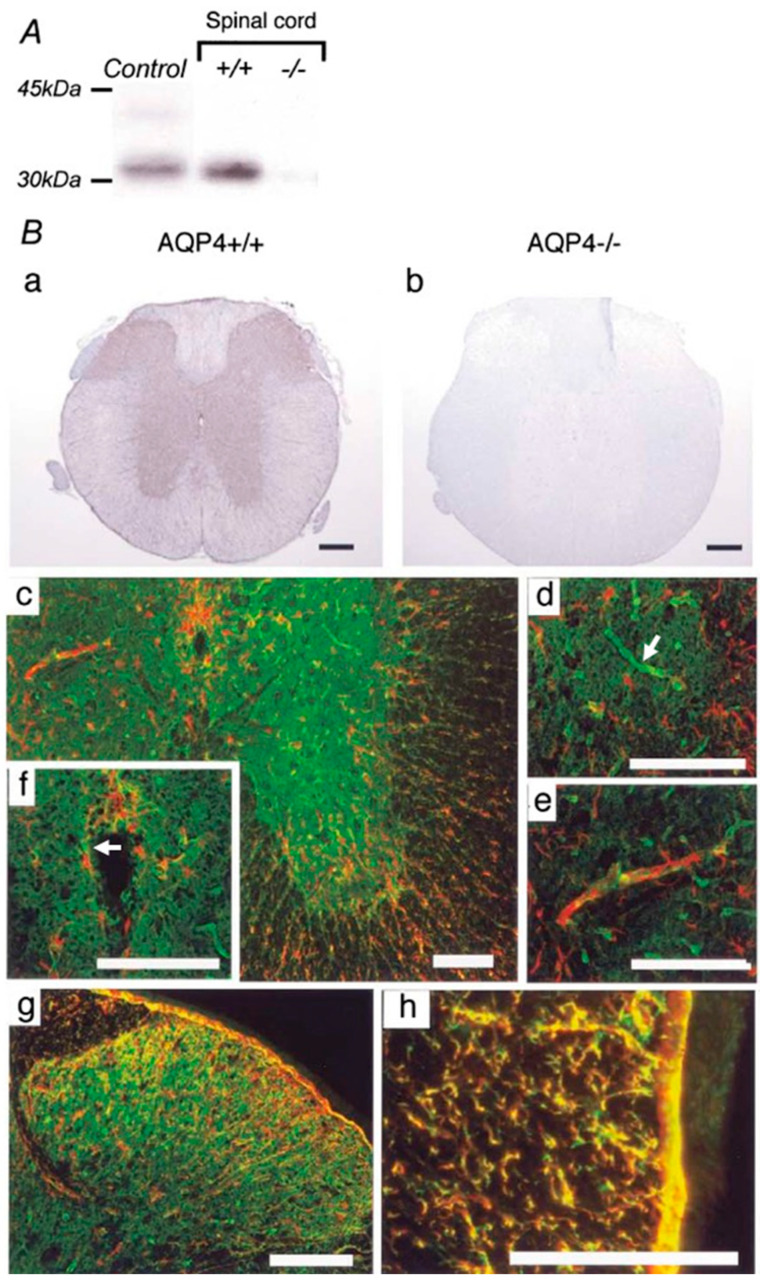
AQP4 expression in mouse spinal cord. (**A**) Western blot analysis demonstrates an approximately 32 kDa band in control cerebral cortex (Control) and wild-type spinal cord (+/+). No expression was detected in the spinal cords of AQP4-deficient mice (^−/−^). (**B**) Immunohistochemistry for AQP4 reveals extensive expression in gray and white matter in the cervical spinal cords of wild-type mice (+/+) (**a**). No specific immunostaining was found in the spinal cords of AQP4 ^−/−^ mice (^−/−^). (**b**). Dense AQP4 staining (green) appeared extensively in gray matter (**c**), especially in astrocytic endfeet surrounding capillaries ((**d**), white arrow). GFAP-immunopositive (red) glial processes extending to capillaries were observed (**e**). Faint AQP4 and GFAP staining was detected in ependymal cells lining the central canal ((**f**), white arrow). GFAP and AQP4 were co-localized in fibrous thin astrocytes in the superficial dorsal horn (**g**). In white matter, AQP4 was co-expressed prominently with GFAP-immunoreactive radial fibrous glial processes surrounding the blood vessels and the glia limitans (**h**). ((**c**–**h**); red: GFAP, green: AQP4). Black scale bar = 0.2 mm. White scale bar = 0.1 mm. Figure and figure legend adopted from [49] with permission.

**Figure 3 cells-12-01701-f003:**
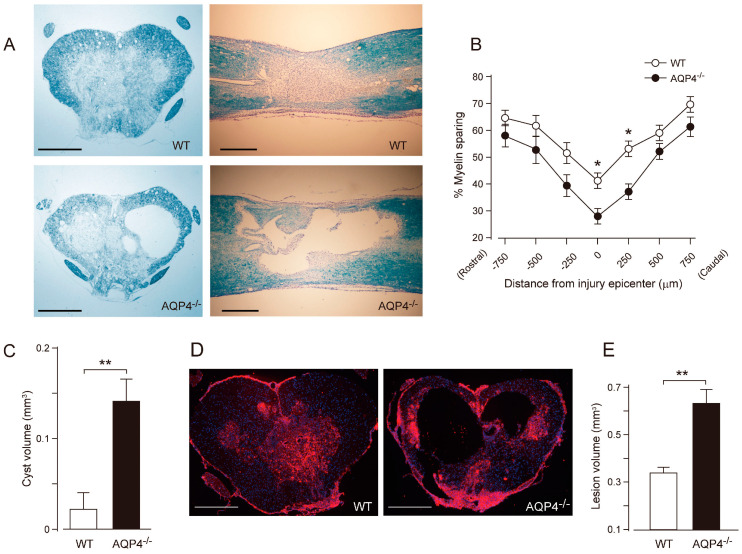
Greater tissue damage and prominent cyst formation in AQP4^−/−^ mice following a thoracic contusion injury. (**A**) Representative images of Luxol Fast Blue (LFB) staining at 42 days post-injury (dpi) (upper left, WT cross section; upper right, WT longitudinal section; lower left, AQP4^−/−^ cross section, lower right; AQP4^−/−^ longitudinal section). Note the greater demyelination and prominent cyst formation in AQP4^−/−^ mice. (**B**) Stereological quantification of LFB staining shows significantly increased myelin loss in AQP4^−/−^ mice. Data are represented as mean ± SEM, *n* = 7 each group; statistical significance was evaluated using a two-way ANOVA with Bonferroni *post-hoc* test, * *p* < 0.05. (**C**) Quantification of cyst volume shows significantly greater cyst volume in AQP4^−/−^ mice (black bar) compared with WT (white bar). Data are represented as mean ± SEM, *n* = 7 each group; statistical comparisons were made using a Student’s *t*-test, ** *p* < 0.01. (**D**) Representative images of fibronectin staining at the injury epicenter (red, fibronectin; blue, DAPI; left, WT; right, AQP4^−/−^). (**E**) Stereological quantification of lesion volume delineated by fibronectin shows significantly greater lesion volume in AQP4^−/−^ mice (black bar) compared with WT mice (white bar). Data are represented as mean ± SEM, *n* = 7 each group; statistical comparisons were made using a Student’s *t*-test, ** *p* < 0.01. Scale bars: A, D, 400 µm. Figure and figure legend adopted from [90] with permission.

**Figure 4 cells-12-01701-f004:**
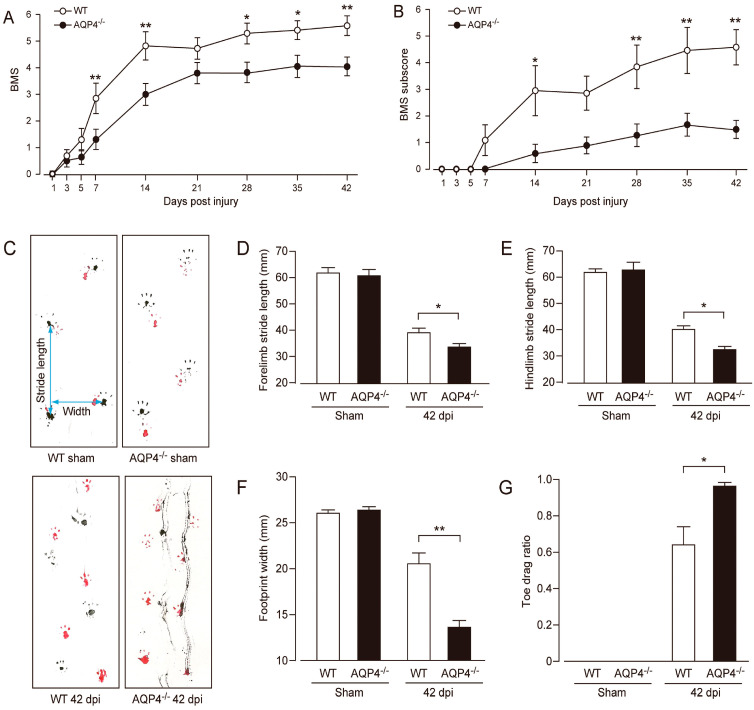
Impaired locomotor recovery following contusion SCI in AQP4^−/−^ mice. AQP4^−/−^ mice had a worse functional outcome as revealed by BMS (**A**) and BMS subscores (**B**) compared with WT mice (white circles, WT; black circles, AQP4^−/−^). Data are represented as mean ± SEM, *n* = 10 each group. Statistical comparisons were made using repeated-measures ANOVA with Bonferroni *post-hoc* test, * *p* < 0.05, ** *p* < 0.01. (**C**) Representative images of footprint analysis (upper left, WT sham; lower left, WT 42 days post-injury, i.e., dpi; upper right, AQP4^−/−^ sham; lower right, AQP4^−/−^ 42 dpi). AQP4^−/−^ mice showed significantly decreased stride length in both forelimbs (**D**) and hindlimbs (**E**), decreased width of hindlimb footprints (**F**), and increased toe drag ratio (**G**) (white bars, WT; black bars, AQP4^−/−^). Data are represented as mean ± SEM, *n* = 10 each group. Statistical significance was evaluated using a Student’s *t*-test or Mann–Whitney *U* test, * *p* < 0.05, ** *p* < 0.01. Figure and figure legend adopted from [90] with permission.

## Data Availability

Not applicable.

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
