# Peer review of "The Role of Aquaporins in Spinal Cord Injury"

_cells, 2023, doi:10.3390/cells12131701_

Round 1
Reviewer 1 Report
This review focuses on the potential role of aquaporin-4 (AQP4) in spinal cord injury. The AQP4 is a major contributor to the edema formation and resolution following spinal cord injury. The authors discuss the critical role of AQP4 in various rodent models and mention some human studies. Impressive and transparent figures help the reader to understand the cellular events following spinal cord injury.
Although the review is thoroughly and very well written there are few points where the MS could be further improved.
1) Glia scar formation depends on reactive astrocytes. Furthermore, the migration of astrocytes to the pathological direction after spinal cord injury is a key step in glial scar formation. The author discuss in the manuscript that AQP4 expression increases in the chronic phase of injury. A short section should be inserted to introduce the relationship between AQP4 and glial scar formation.
2) Neuroinflammation plays a critical role following spinal cord injury. Does the downregulation of AQP4 alleviate neuroinflammatory processes?
Author Response
Thank you for your detailed review of our manuscript.
Based on your suggestion, we have added an entirely new section (section 7) to the manuscript entitled "AQP4, glial scar formation, and neuroinflammation".
Reviewer 2 Report
In this manuscript, Garcia T and co-authors comprehensively and systematically reviewed the critical role of aquaporin-4 (AQP4) water channel in edema formation and resolution after spinal cord injury (SCI), as well as the therapeutic potential of AQP4 modulation in edema resolution and functional recovery. Therefore, the futher studies focusing on elucidating the expression and subcellular localization of AQP4 during specific phases after SCI will inform therapeutic modulation of AQP4 for optimization of histological and neurological outcome.
I only have one minor suggestion.
In Figure 1 legend, Lines 105-106: "Water selectivity depends on two pore helices and their highly conserved Asn–Pro–Ala motifs." Here, the authors should indicate the specific amino acid residues located in the ar/R region that decides the water selectivity.
Author Response
Thank you for your detailed review of our manuscript.
We have changed Figure 1 to a new figure which shows the water selectivity of the AQP4 channel. AQP4 assembles as a tetramer with each monomer comprising six transmembrane helices and two half-helices. The two half helices harbor the aquaporin signature motif (NPA) as well as part of the aromatic-arginine (ar/R) motif that functions as a selectivity filter for water molecules.
Round 2
Reviewer 1 Report
The authors answered the questions satisfactorily. The manuscript is acceptable for publication in its present form.
Author Response
Thanks to Reviewer 1 for helpful comments.